# KNOWLEDGE DEBUGGER: DIAGNOSIS OF KNOWLEDGE INCONSISTENCY WITH MULTIMODAL GRAPH

## ABSTRACT

Human knowledge can naturally be organized as multimodal graphs, with prime examples including research papers or Wikipedia pages. However, identifying information inconsistencies within such knowledge-intensive documents remains challenging. These inconsistencies can be explicit, such as numerical discrepancies between tables and their textual descriptions, or implicit, like differing conclusions presented at the beginning and end of an article. Large Language Models (LLMs) have shown great potential in detecting these types of inconsistencies. Nevertheless, their practical deployment is often hindered by limitations such as restricted context windows and high inference costs. Additionally, standard Retrieval-Augmented Generation (RAG) approaches struggle to effectively capture intricate reference relationships within multimodal graphs. To address these challenges, we propose Knowledge Debugger, an efficient Graph Neural Network (GNN)-based framework that can identify diverse types of knowledge inconsistencies in multimodal data. To evaluate the effectiveness of our method, we built a Multimodal Knowledge Debugging Benchmark (MKDB) including 3 modalities, 699 Wikipedia pages, more than 10000 research papers, and more than 10000 knowledge-debugging tasks with answers. With our approach, we leverage LLMs to generate high-quality labels for training multimodal GNNs. The trained GNNs demonstrate strong performance in consistency checking tasks on multimodal graphs. Specifically, we beat the best RAG methods by 11% on node-level bug detection tasks. By employing GNNs, we significantly enhance system efficiency and scalability, enabling effective and practical inconsistency detection in complex multimodal knowledge structures.

## 1 INTRODUCTION

Human knowledge inherently possesses a multimodal structure where dense information is organized across multiple modalities with complex relational structures. Prime examples include knowledge-intensive sources such as Wikipedia pages Wikipedia (2025) and academic papers Kinney et al. (2023) with rich text, figures, tables, and numerous hyperlinks and cross-references. Other knowledge-intensive platforms such as autonomous vehicles Cui et al. (2023); Xiao et al. (2022), healthcare Yildirim et al. (2024); Krones et al. (2024), customer support automation Liao et al. (2018); Saha et al. (2018), and anomaly detection Wang et al. (2023); Liu et al. (2025a), *etc*, are also highly structured and multimodal. To ensure the effectiveness of this knowledge, a key research question arises: *Can we efficiently detect knowledge inconsistency within structured multimodal data?*

Inconsistencies, while undesirable, are ubiquitous in knowledge databases. For instance, Wikipedia pages often contain partially updated information, where certain sections reflect recent changes while others still retain outdated content; in academic writing, researchers frequently struggle to ensure consistency across statements in long-context, multimodal, and cross-referenced content; in safety-critical systems, *e.g.*, autonomous driving and healthcare, where inconsistencies can arise due to malicious intent or naturally induced distribution shifts; these inconsistencies, if not identified, can lead to dire consequences Zhao et al. (2024).

However, detecting knowledge inconsistency is a complex and challenging task, especially when the data is multimodal and structured. First, there often lack ground-truth labels for inconsistencies, making training and evaluations challenging. Second, inconsistencies are often subtle, where a statement may appear plausible on its own, yet its inconsistency can only be detected when considering

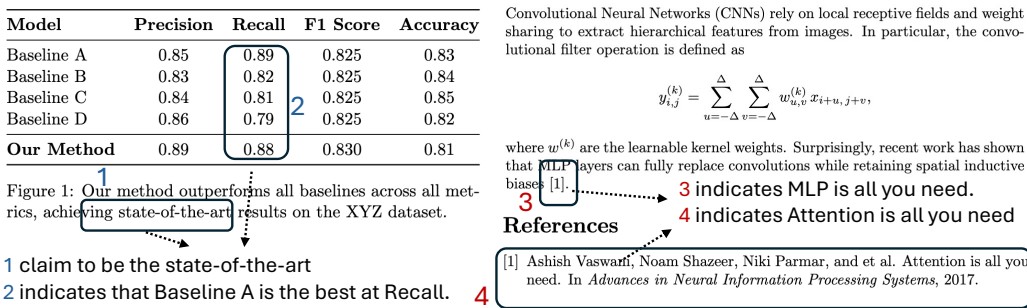

Figure 1: **Identifying inconsistencies in multimodal and structured content such as research papers and Wikipedia pages is challenging, even for humans.** Left: a claim about the method is inconsistent with the information in the table, constituting a node-level inconsistency across 2 modalities. Right: the citation of "Attention is all you need" is inconsistent with a claim about MLP in the paper, constituting an edge-level in consistency across papers.

the (ultra-)long context with its potential relational structure holistically. Lastly, while state-of-the-art multimodal large language models (LLMs), *e.g.*, GPT-4o OpenAI et al. (2024) or Claude 3.7 Cla, may be used to diagnose multimodal content, the associated token cost could be prohibitive. For instance, identifying reference inconsistencies in one paper requires inputting the full content of multiple papers, which will exceed millions of tokens in a short time.

There are existing datasets and solutions that can be used for knowledge inconsistency detection, but they have limitations in several aspects. First, although benchmarks on multimodal data, such as WikiWeb2M on Wikipedia Burns et al. (2023a) exist, they focus on Question Answering tasks, which can usually be answered well with strong LLMs that have sufficient external knowledge. Second, there are existing solutions that can be used for knowledge inconsistency detection, but they are often limited in effectiveness and/or efficiency. For example, retrieval-augmented generation (RAG) models have been used to detect inconsistencies; however, these methods are solely based on the semantic similarity between the content, without considering the relational information across the content Lewis et al. (2021). There currently lacks a comprehensive benchmark for multi-modal knowledge inconsistency detection, and a detection algorithm that can fully leverage the relational information among data, necessary for detecting subtle inconsistencies within a knowledge base.

In this paper, we propose Knowledge Debugger, a Graph Neural Network (GNN) based framework designed for efficient training and inference in multi-modal knowledge inconsistency detection. Our key insight is that multimodal knowledge-intensive data are naturally suited to graph-based representations with entities, concepts, and their relations as nodes and edges. Moreover, we can formulate inconsistency detection tasks as node classification tasks, where the content exhibits inconsistency, or edge classification tasks, where there are erroneous relations among entities. At its core, Knowledge Debugger trains a graph neural network-based retriever that learns to retrieve structurally and contextually relevant content, enabling robust inconsistency detection across both modalities and document relations.

To demonstrate the effectiveness of Knowledge Debugger, we construct a novel benchmark dataset named as Multimodal Knowledge Debugging Benchmark (MKDB) that captures fine-grained knowledge inconsistencies within Wikipedia articles and research papers. In this benchmark, we include 3 modalities (text, table, and image), more than 10,000 research papers, 699 Wikipedia pages, and more than 10,000 knowledge debugging tasks. We introduce inconsistencies where different parts of a document may appear semantically similar yet contain conflicting factual information, highlighting the subtle and often overlooked nature of such inconsistencies. Concretely, we introduce four types of debugging tasks in MKDB to simulate real-world knowledge debugging tasks: (1) node-level bug detection, (2) node-level bug correction (3) edge-level bug detection, (4) edge-level bug correction.

Empirical results demonstrate that our approach outperforms traditional similarity-based baselines, particularly in identifying fine-grained inconsistencies that prior methods fail to capture. Based on our experimental results on our benchmark, our proposed GNN-based algorithm outperforms the best RAG-based method by 11% while remaining efficient.

## 2 RELATED WORKS

**Factual inconsistency detection**. Factual inconsistency is a critical challenge in various natural language processing tasks. Traditional approaches often relied on training entailment or classification models to detect contradictions between generated text and source knowledge Cao et al. (2018); Kryściński et al. (2019); Tang et al. (2022). Question answering has also been explored as a means to evaluate consistency Durmus et al. (2020). Furthermore, researchers have the use of retrieval-augmented generation (RAG) methods, where models dynamically retrieve external factual data to solve the hallucination and inconsistency.Jiang et al. (2023); Ma et al. (2023) Recent efforts in this domain explore the use of LLMs for evidence retrieval and claim verification, often incorporating techniques like chain-of-thought reasoning to improve performance Kojima et al. (2022). Notably, evaluating factual inconsistencies has been an open research question. Benchmarks like FEVER have been instrumental in evaluating systems' ability to verify claims against evidence Thorne et al. (2018), and more specialized datasets like SciFact focus on scientific claim verification Wadden et al. (2022).

**Multimodal knowledge graph**. Multimodal knowledge graphs offer a powerful way to represent structured and unstructured information from diverse sources, such as webpages and academic research papers Burns et al. (2023a); Galiano et al. (2023). These graphs integrate various modalities, including text, tables, figures, and metadata into a unified representation, enabling a more comprehensive understanding of the underlying knowledge Chen et al. (2023). In the context of scientific articles, multimodal knowledge graphs can capture both structured information, such as citations and references, and unstructured content, including the full text and visual elements Zhang et al. (2023). Recent advancements in this field have seen the application of LLMs for extracting entities and relations from both textual and visual modalities within documents Lee et al. (2024). For instance, LLMs can be used to understand the content of figures and their captions, linking them to relevant parts of the text and other entities in the graph Liu et al. (2025b). Platforms are emerging that leverage multimodal knowledge graphs to align different components of research papers, including text, diagrams, and even code, facilitating complex queries and discovery of knowledge across modalities Kannan et al. (2020). Similarly, in the context of Wikipedia, multimodal knowledge graphs combine article text with information from infoboxes and images, enriching the graph with contextual information Yoon et al. (2023). These knowledge graphs support advanced applications in question answering, fact-checking, and scientific discovery, allowing systems to retrieve and reason over diverse types of evidence Yao et al. (2023). Researchers are increasingly exploring the use of LLMs not only to build but also reason over multimodal knowledge graphs, enabling more sophisticated information retrieval and inference capabilities Pan et al. (2024).

## 3 GRAPH STRUCTURE OF MULTIMODAL DATA

In knowledge-intensive and multimodal knowledge sources, *e.g.*, Wikipedia articles and scientific papers, identifying knowledge inconsistency within one modality or cross-modality is a significant challenge as these sources contain implicit yet meaningful relationships across different content modalities. These relationships are not purely semantic; rather, they emerge from the spatial arrangement, document flow, and explicit cross-references within the source. To capture this, we model such content using a multimodal graph, where each unit—text, image, table, or metadata is represented as a node, and edges encode the spatial, referential, or sequential dependencies across these units.

**Multimodal graph.** We define a multimodal graph $G = (\mathcal{V}, \mathcal{E}, M, L)$, where $\mathcal{V}$ is the node set, $\mathcal{E} \subseteq \mathcal{V} \times \mathcal{V}$ the directed edges, $M : \mathcal{V} \to \mathcal{T} \cup \mathcal{I} \cup \mathcal{B}$ maps each node to a text/image/table attribute, and $L : \mathcal{E} \to \mathcal{R}$ labels edges with relation types. We instantiate $\mathcal{R}$ to reflect human reading flow—*next* (sequential order within sections, e.g., paragraph → paragraph or text → caption), *reference* (citing span → cited item, e.g., "Fig. 2" or "[12]"), and *follow* (claim/mention → its immediate elaboration/result)—so we target inconsistencies within a reader's local attention range. Richer cross-paragraph or logical relations can be added as extensions via $L$, but they typically require heavier construction pipelines and are noisier.

**Example: multimodal graph for Wikipedia.** For Wikipedia pages, we focus on two primary modalities: text and images. Each paragraph in a given article is treated as a text node ($M(v) = \mathcal{T}$), while each embedded image is represented as an image node ($M(v) = \mathcal{I}$). These node types arise naturally from how Wikipedia articles are written where textual descriptions are often supported by

relevant visual media, and each content block can be cleanly isolated into a graph node. We define two edge types to model the relationships between these nodes. First, *Follow* edges (Follow$(v_i, v_j)$) capture the sequential ordering in an article where the ordering could be defined between adjacent text paragraphs or between adjacent images. Second, *Reference* edges (Reference$(v_i, v_j)$) are introduced when a paragraph references a nearby image (as shown in Figure 3) or when an image is spatially adjacent to a paragraph and helps illustrate its content. These relationships allow the graph to reflect both the document's layout and its implicit multimodal dependencies.

**Example: multimodal graph for academic papers.**
Scientific papers contain richer multimodal structures. We decompose each paper into three types of nodes: text nodes ($\mathcal{T}$) for paragraphs, section headings, and captions; image nodes ($\mathcal{I}$) for figures and visual diagrams; table nodes ($\mathcal{B}$) for tabular data. These distinctions emerge from the functional roles that each modality plays in conveying scientific knowledge—tables often summarize results, and figures illustrate models or findings. The graph of a paper contains several structurally motivated edge types. Like Wikipedia, Follow edges (Follow) connect adjacent nodes based on document order, enabling flow-based reasoning across sections. Reference edges (Reference) link paragraphs to figures or tables they mention (e.g., "see Table 2"), capturing the interplay between textual explanation and visual evidence. Finally, *Cite* edges (Cite$(v_{\text{text}}, v_{\text{ref}})$) connect text passages to the references they cite, modeling scholarly attribution and inter-document dependency.

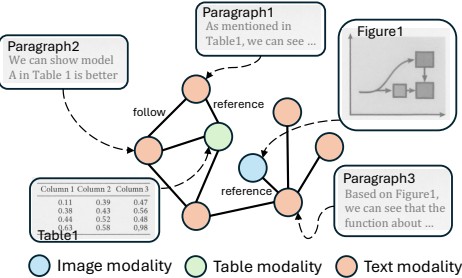

Figure 2: **Research papers or Wikipedia pages can be formed as multimodal graphs.** Each node can represent part of the paper, and each edge can represent following, including, referencing relationships that naturally exist in the paper. Multiple papers are connected through citation edges.

## 4 MULTIMODAL KNOWLEDGE INCONSISTENCY

In a multimodal graph, where heterogeneous content units (e.g., text, images, tables) are interconnected to represent structured knowledge, *inconsistency* refers to a structural bug that violates the internal coherence of the graph. We define **multimodal knowledge inconsistency** as a contradiction, misalignment, or erroneous linkage that disrupts the factual or logical integrity of the information encoded in the graph. Such inconsistencies occur when the content of a node is incompatible with information elsewhere in the graph, or when connections between nodes are semantically invalid.

Formally, given a multimodal graph $G = (\mathcal{V}, \mathcal{E}, M)$, inconsistencies can take two primary forms:

**Node-level inconsistency**. A node $v_i \in \mathcal{V}$ is said to be inconsistent if its content contradicts or conflicts with that of one or more other nodes $\{v_j\}_{j \in \mathcal{N}_i}$, where $\mathcal{N}_i \subseteq \mathcal{V} \setminus \{v_i\}$. Examples include contradictory claims between paragraphs, inconsistent numerical values between a text and a table, or a caption that misinterprets an associated figure.

**Edge-level inconsistency**. An edge $e_{i,j} \in E$ is inconsistent if it represents an incorrect or misleading relationship, such as an invalid reference, an erroneous citation, or a structural link that falsely implies semantic relevance. These edges should not exist in a logically coherent graph.

These inconsistencies may be subtle and span multiple modalities, as shown in Figure 1, making them difficult to detect using unimodal or surface-level similarity methods. For instance, a table may provide updated statistics that conflict with outdated textual claims, or an image may visually contradict a paragraph that refers to it. The inconsistency of multimodal knowledge thus represents a fundamental flaw—akin to a *bug*—in the structure of the graph. Identifying and correcting such inconsistencies is crucial for enabling trustworthy multimodal reasoning and improving downstream tasks such as fact verification, summarization, and knowledge retrieval.

## 5 BUILDING KNOWLEDGE DEBUGGER WITH GNNS

To detect and repair structural bugs in multimodal graphs, we introduce a Graph Neural Network (GNN)-based approach named Knowledge Debugger, operating at both node and edge levels. We

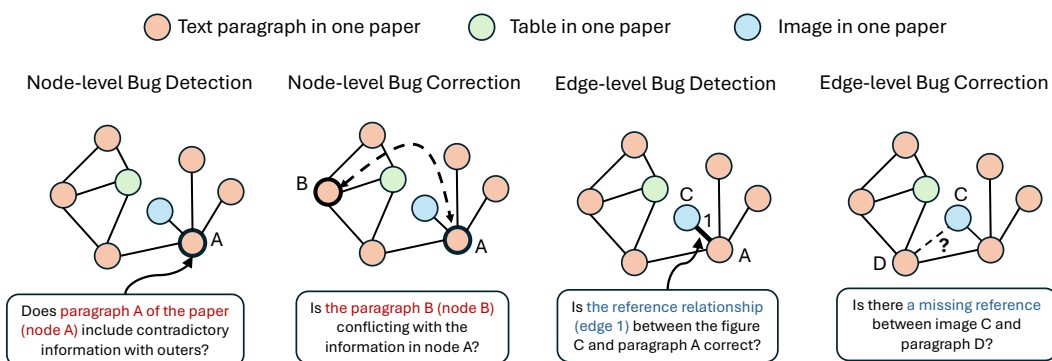

Figure 3: **Four types of knowledge debugging tasks defined based on multimodal graphs.** Only the first type of knowledge debugging tasks is considered as the node classification tasks; the remaining three types of debugging tasks can be understood as special forms of edge classification.

frame the debugging process in two distinct stages: (1) *bug detection*—identifying inconsistencies and determining their types, and (2) *bug correction*—localizing and resolving these inconsistencies. Our approach leverages the semantic information of multimodal content alongside the structural context inherent in the graph topology, utilizing specialized models tailored for graph modeling tasks.

**Semantic and structural representation in GNNs**   Capturing semantic context is crucial for identifying inconsistencies in knowledge-intensive domains. Initially, we set each node's hidden state using embeddings from a pretrained multimodal encoder:

$$\mathbf{h}_i^{(0)} = \text{MENC}(v_i) \tag{1}$$

where $\text{MENC}(\cdot)$ denotes a modality-specific embedding extractor (*e.g.*, voyage-multimodal-v3 Voyage AI (2024)). These multimodal embeddings encode semantic content, facilitating meaningful message propagation in subsequent stages. At each GNN layer $l$, node hidden states are updated by aggregating messages from neighboring nodes:

$$\mathbf{h}_i^{(l)} = \text{AGG}\left(\mathbf{h}_i^{(l-1)}, \left\{\text{MSG}\left(\mathbf{h}_j^{(l-1)}\right) : v_j \in \mathcal{N}(v_i)\right\}\right) \tag{2}$$

Here, $\text{MSG}(\cdot)$ computes messages based on neighbor states, and $\text{AGG}(\cdot)$ combines these messages with the node's previous state. After $L$ layers, the final hidden state $\mathbf{h}_i^{(L)}$ captures a rich combination of semantic and structural context suitable for downstream debugging tasks.

**Training node-level debugger**   At the node level, debugging involves two subtasks: (1) *bug detection*—deciding if a node $v_i$ has inconsistent information, and (2) *bug correction*—identifying other nodes that conflict with the target node. For detection, given $v_i$ in graph $G$, we train a binary classifier $f_\theta(v_i; G) \in \{0, 1\}$ with ground-truth consistency labels $y_i$. For correction, once an inconsistency at $v_i$ is detected, we further identify relevant supporting nodes by classifying candidates $v_j \in \mathcal{N}(v_i)$ using another binary classifier $f_\phi^{\text{loc}}(v_i, v_j) \in \{0, 1\}$; this is essentially an edge-classification task between the bug node $v_i$ and remaining candidate nodes $v_j$. Concretely, we minimize

$$\mathcal{L}_{\text{det}}(\theta) = \sum_{v_i \in \mathcal{V}_{\text{train}}} \mathcal{L}_{\text{BCE}}(f_\theta(v_i), y_i) \qquad \mathcal{L}_{\text{corr}}(\phi) = \sum_{(v_i, v_j) \in \mathcal{P}} \mathcal{L}_{\text{BCE}}(f_\phi^{\text{loc}}(v_i, v_j), z_{ij}) \tag{3}$$

where $z_{ij} = 1$ indicates a conflict between $v_i$ and $v_j$ that requires resolution by adjusting the information within these nodes. Based on the classification scores over $v_j$, Knowledge Debugger can be conveniently combined with RAG systems to resolve the bug in $v_i$ using the top-$K$ supporting nodes.

**Training edge-level debugger**   Edge-level debugging similarly involves two subtasks: (1) *bug detection*—determining if an edge $(v_i, v_j)$ is invalid, and (2) *bug correction*—suggesting correct alternative target nodes. This debugging process checks whether the connected information is

accurately referenced or cited. For detection, we train an edge classifier $f_\theta^{\text{edge}}(v_i, v_j) \in \{0, 1\}$; for correction, we train another edge classifier $f_\psi^{\text{corr}}(v_i, v_j) \in \{0, 1\}$ to suggest proper connections. Concretely, we minimize

$$\mathcal{L}_{\text{det}}^{\text{edge}}(\theta) = \sum_{(v_i, v_j) \in \mathcal{E}_{\text{train}}} \mathcal{L}_{\text{BCE}}\big(f_\theta^{\text{edge}}(v_i, v_j),\, y_{ij}\big) \qquad \mathcal{L}_{\text{corr}}^{\text{edge}}(\psi) = \sum_{(v_i, v_j) \in \mathcal{P}} \mathcal{L}_{\text{BCE}}\big(f_\psi^{\text{corr}}(v_i, v_j),\, z_{ij}\big)$$

(4)

where $y_{ij}$ denotes edge validity and $z_{ij} = 1$ if $v_j$ is a correct node to which $v_i$ should connect. Notably, edge-level bug correction is often more convenient than node-level correction, since the predicted proper connections from $f^{\text{corr}}(\cdot)$ can be directly taken as the corrected edges without relying on additional RAG systems.

**Inference procedure for knowledge debugging** Both node-level and edge-level models follow a two-stage inference pipeline: (1) *Detection*, where models identify inconsistencies based on supervised signals; and (2) *Correction*, where detected inconsistencies are localized and corrected using contrastive or ranking-based objectives. By integrating semantic embeddings and graph-aware reasoning, Knowledge Debugger effectively identifies and corrects subtle multimodal inconsistencies, enhancing robustness in knowledge verification and repair tasks.

# 6 MULTIMODAL KNOWLEDGE DEBUGGING BENCHMARK (MKDB)

To systematically evaluate Knowledge Debugger, we construct a comprehensive benchmark comprising two representative types of knowledge-intensive materials: Wikipedia articles and research papers. Within this benchmark, we include two types of tasks mentioned in Section §5 *i.e.*, bug detection and bug correction. In this section, we provide details about the benchmark as well as its construction process and evaluation metrics.

## 6.1 DATA COLLECTION AND GRAPH CONSTRUCTION

**Benchmark corpus details**. Table 1, shows the size and scale of our multimodal benchmark dataset and constructed knowledge debugging tasks. In our benchmark, we provide two types of data: (i) multimodal data source with constructed multimodal graphs from research papers and Wikipedia pages; and (ii) knowledge debugging tasks with questions and answers for both node-level and edge-level bug detection and bug correction. For multi-modalities, for the Wikipedia pages, we have 700 image nodes, 7,776 text nodes. For the research papers, we have 1,402,305 text nodes, 30,812 image nodes, and 33,024 table nodes.

**Collection of multimodal knowledge**. To construct multimodal graphs and build tasks on the edges and nodes, we require fine-grained, high-quality data from both Wikipedia and research papers. For Wikipedia, we utilize the WikiWeb2M Burns et al. (2023b) dataset, which provides paragraph-level splitting of text together with its surrounding words. For research papers, however, no comparable dataset exists, so we self-collect 10,000 computer science papers from arXiv [1] , parsing and preprocessing the source files along with their figures and tables to obtain texts, tables, images, citations, and metadata.

**Graph construction with multimodal knowledge**. Building on these resources, the MKDB benchmark is constructed through several steps: (1) we collect raw content from Wikipedia and arXiv papers; (2) we parse this content into nodes such as paragraphs, images, and captions using rule-based methods; (3) we add edges based on reading order and reference links; and (4) we connect sections and subsections with their corresponding content to capture structural coherence. Importantly, no content is altered during this process—ensuring full reversibility to the original material—and we rely solely on rule-based preprocessing without additional annotation or external supervision. Since both Wikipedia articles and research papers are generally high-quality and knowledge-dense, we expect relatively few inherent inconsistencies in the raw sources.

---

[1]https://arxiv.org/

Table 1: **Dataset and benchmark statistics.** Left: statistics of multimodal data sources (#Documents indicates the number of raw documents; #Node and #Edge indicate the constructed graph structure). Right: size of the proposed MKDB benchmark (number of datapoints for node/edge-level tasks). Each node-level or edge-level task can be converted to a bug detection and a bug correction sub-task.

|  | #Documents | #Node | #Edge |  | Node-level task | Edge-level task |
|---|---|---|---|---|---|---|
| Wikipedia | 700 | 8,476 | 11,865 | Wikipedia | 4,987 | 4,381 |
| Research | 10,393 | 1,466,141 | 4,804,388 | Research | 4,567 | 5,003 |

## 6.2 TASK AND EVALUATION DESIGN

**Construction of edge-debugging tasks.** For a fixed node $u$, we generate edge-level bugs by replacing its entire set of neighbors with new nodes drawn from a weighted neighborhood distribution. Concretely, for each original neighbor $v \in \mathcal{N}(u)$, we sample a replacement $v' \in \mathcal{N}^{(K)}(u) \setminus \mathcal{N}(u)$ according to

$$q_u(x) = \frac{w(u, x)}{\sum_{y \in \mathcal{N}^{(K)}(u) \setminus \mathcal{N}(u)} w(u, y)}, \tag{5}$$

where $w(u, x)$ is a weight function (e.g., based on positional distance or semantic similarity). The corrupted neighborhood $\tilde{\mathcal{N}}(u) = \{v'_1, \ldots, v'_{d(u)}\}$ then replaces the original $\mathcal{N}(u)$, yielding edges $(u, v'_i)$ that preserve local degree but induce controlled inconsistencies.

**Construction of node-debugging tasks**. For a fixed node $u$, we also apply weighted neighborhood sampling, but instead of rewiring edges, we perturb its attributes. Specifically, for each sampled neighbor, we prompt an LLM to generate conflicting information, which is then injected back into $u$'s attributes. This produces a perturbed representation of $u$ that intentionally conflicts with multiple related nodes, yielding realistic node-level inconsistencies.

**Evaluation metrics**. Since bug detection and correction address different stages of knowledge debugging, we evaluate them with tailored metrics: detection is assessed using binary classification measures (precision, recall, and F1), while correction is treated as a retrieval task and judged by ranking metrics (Recall@k, MRR, MAP, and NDCG@k) to capture the quality of the top candidates.

## 7 EXPERIMENTAL SETTING

**Baseline settings**. To comprehensively evaluate our proposed methods, we compare against several baselines that utilize large language models (LLMs). Given their robust in-context learning capabilities, LLMs are expected to detect inconsistencies within textual domains automatically. In addition to the standard Retrieval-Augmented Generation (RAG) approaches that leverage either text-based or multimodal embeddings, we explore retrievers based on graph structure, specifically employing personalized PageRank (PPR). Furthermore, to thoroughly assess model capabilities, we introduce hybrid retrievers that combine scores from text-based retrievers and PPR-based graph structure retrievers through a linear weighting scheme. All experimental results reported in Table 2 and Table 3 are derived from a fixed test set of 400 examples within the knowledge debugging benchmark separately on the wiki graph and the paper graph. Notably, retrievers and LLMs are utilized without further training, while the GNN model is specifically trained on the remainder of the benchmark data.

**Model settings**. In our retrieval-augmented generation (RAG) experiments, we select Qwen2.5-7B-Instruct-Turbo [1] as the foundational language model. For textual retrieval, we employ the sentence transformer all-MiniLM-L6-v2 [2] as a benchmark. For multimodal retrieval scenarios, we utilize voyage-multimodal-3 [3] and AltCLIP (Chen et al., 2022), both of which are recognized for achieving state-of-the-art performance in multimodal understanding tasks. The graph neural network component in our experiments is implemented using `GATv2Conv`. Additional technical details and experimental parameters are elaborated upon in the Appendix.

---

[1]We utilize Qwen2.5-7B-Instruct-Turbo from TogetherAI: https://huggingface.co/Qwen/Qwen2.5-7B-Instruct
[2]https://huggingface.co/sentence-transformers/all-MiniLM-L6-v2
[3]https://blog.voyageai.com/2024/11/12/voyage-multimodal-3/

Table 2: **MKDB benchmark results on bug detection tasks**. RAG (oracle) indicates that we directly provide the ground-truth node to the LLM for generation. Additionally, RAG (hybrid retriever) indicates that this retriever utilizes a combination of a text retriever and a structural retriever.

| | Node-level | | | Edge-level | | |
|---|---|---|---|---|---|---|
| Method | Precision ↑ | Recall ↑ | F1 ↑ | Precision ↑ | Recall ↑ | F1 ↑ |
| **Research** | | | | | | |
| RAG (oracle) | 82.3 | 95.0 | 88.2 | — | — | — |
| LLM (w/o retrieval) | 65.3 | 55.5 | 60.0 | 53.1 | 21.5 | 30.6 |
| RAG (structural retrieval) | 66.5 | 60.5 | 63.4 | 4.2 | 2.0 | 2.7 |
| RAG (text retrieval) | 81.8 | 40.5 | 54.2 | 50.3 | 41.0 | 45.2 |
| RAG (multimodal retrieval) | 66.3 | 55.0 | 60.1 | 57.6 | 57.2 | 57.4 |
| RAG (hybrid retrieval) | 77.4 | 56.5 | 65.3 | 46.6 | 45.0 | 45.8 |
| **Wikipedia** | | | | | | |
| RAG (oracle) | 69.3 | 95.0 | 80.2 | — | — | — |
| LLM (w/o retrieval) | 68.2 | 67.5 | 67.8 | 59.9 | 48.5 | 53.6 |
| RAG (structural retrieval) | 64.6 | 52.0 | 57.6 | 52.9 | 18.5 | 27.4 |
| RAG (text retrieval) | 68.2 | 87.0 | 76.5 | 51.4 | 64.5 | 57.2 |
| RAG (multimodal retrieval) | 65.1 | 82.0 | 72.6 | 54.2 | 74.5 | 62.7 |
| RAG (hybrid retrieval) | 69.8 | 90.0 | 78.6 | 53.2 | 50.0 | 51.6 |
| Ours (GNN-based) | 84.1 | 90.0 | 87.0 | 57.7 | 82.1 | 67.8 |

# 8 EXPERIMENTAL RESULTS

We conduct comprehensive experiments on both the paper and wikipedia graphs for node-level and edge-level knowledge debugging. Our main findings are:

**Knowledge Debugger outperforms LLM-based methods on both detection and correction tasks**. As shown in Table 2, our GNN-based approach achieves the highest F1 scores on both node-level and edge-level bug detection. In particular, on node-level detection, it attains an F1 of 0.870—nearly matches the performance of the LLM with an oracle retriever. This improvement stems from the GNN's ability to model structural perturbations: when the graph structure changes, the semantic relationships among document components shift, and a GNN can learn to propagate and detect these inconsistencies across the graph.

**Similarity-based retrieval alone fails to spot subtle inconsistencies**. Our benchmark is designed so that knowledge inconsistencies are often subtle and require careful reading to detect. Consequently, even state-of-the-art text-based and multimodal retrievers fall short, as evidenced by their low scores in Table 2. Simple similarity measures cannot reliably capture the nuanced contradictions that occur within a single document graph.

**Hybrid retrievers that combine structure and semantics yield better results**. Table 2 also shows that hybrid retrieval—integrating both text-based and structure-based signals—consistently outperforms methods that rely solely on one or the other. This demonstrates that leveraging structural context alongside semantic similarity is crucial for effectively identifying knowledge inconsistencies in both paper and wiki graphs.

# 9 DISCUSSION

**RQ1: What does the Knowledge Debugger learn during training?** Each node in our multimodal graph is initialized using embeddings from state-of-the-art models such as voyage-multimodal-v3. Consequently, at the start of training, our Graph Neural Network (GNN) primarily leverages semantic similarity between nodes for message passing. Throughout the training process, the GNN learns to discern semantic similarity structures within subgraphs, particularly identifying when nodes present conflicting information or when nodes maintain consistent and coherent information within a subgraph. This supervision enables the GNN to recognize and internalize patterns associated with "buggy" nodes. Our experiments further reveal that even when applying only scalar-based similarity

Table 3: **MKDB benchmark results on bug correction tasks**. R@5 represents Recall@5. The correction task directly utilizes the retriever without asking LLMs for an answer.

| Method | Node-level | | | | Edge-level | | | |
|---|---|---|---|---|---|---|---|---|
| | MRR ↑ | MAP ↑ | R@5 ↑ | NDCG@5 ↑ | MRR ↑ | MAP ↑ | R@5 ↑ | NDCG@5 ↑ |
| | | | | **Research** | | | | |
| Structure | 22.9 | 12.6 | 21.4 | 16.5 | 95.8 | 74.5 | 80.0 | 83.3 |
| Text | 34.4 | 14.7 | 19.0 | 19.2 | 84.1 | 54.5 | 61.8 | 66.7 |
| Multimodal | 13.7 | 5.5 | 8.6 | 6.9 | 23.8 | 8.3 | 13.5 | 13.9 |
| Hybrid | 26.3 | 15.0 | 24.3 | 19.0 | 84.5 | 73.4 | 85.5 | 80.7 |
| | | | | **Wikipedia** | | | | |
| Structure | 73.8 | 45.9 | 49.8 | 52.1 | 40.8 | 18.1 | 19.9 | 23.4 |
| Text | 85.8 | 50.1 | 53.4 | 57.8 | 66.8 | 53.6 | 62.8 | 56.7 |
| Multimodal | 87.3 | 57.5 | 60.7 | 64.8 | 70.3 | 62.8 | 79.1 | 67.5 |
| Hybrid | 86.3 | 50.8 | 54.4 | 58.9 | 68.0 | 53.2 | 60.6 | 56.8 |
| Ours | 81.8 | 60.8 | 60.5 | 61.8 | 89.2 | 85.6 | 99.1 | 90.2 |

scores for nodes, the GNN effectively identifies graph structures indicative of problematic nodes. Furthermore, employing higher-dimensional features contributes to more stable training.

**RQ2: How cheap and efficient is the Knowledge Debugger?** Compared to LLM baselines, our GNN-based Knowledge Debugger is *far smaller* and *substantially faster*, enabling efficient batch inference at scale. Concretely, we use a 3-layer `GATv2Conv` encoder with 4 attention heads and 128 hidden units, yielding <10M parameters—orders of magnitude below the 7B-parameter LLMs used in prior baselines. This compact model fits easily on a single commodity GPU and supports high-throughput, paragraph-level inference, which is critical for large-scale bug scanning across papers and wiki pages. In contrast, running an LLM on every paragraph would incur prohibitive latency and cost, making end-to-end scanning impractical; the GNN therefore provides a practical, cost-effective backbone for this workload.

## 10 ABLATION STUDY

**Embedding initialization.** To assess the importance of semantic initialization, we replaced the initial hidden states provided by voyage-multimodal-3 embeddings with randomly initialized hidden states. After training, the model achieved an F1 score around 0.5 or lower for edge classification, significantly below our best results. This clearly indicates that semantic initialization plays a crucial role in enabling our Knowledge Debugger to identify buggy nodes effectively.

**GNN layer number.** We experimented with different numbers of layers in the GNN architecture to determine their influence on performance. With only 1 layer, the model reached an F1 score of 0.5845, which improved slightly to 0.5959 with 2 layers. The performance peaked at 0.6957 with 3 layers but decreased to 0.6570 when we further extended to 4 layers. This suggests that adding more layers beyond a certain point does not provide additional performance benefits.

**GNN backbone.** Lastly, we tested the impact of various GNN backbone architectures on the overall performance. Replacing our chosen `GATv2Conv` backbone, we experimented with standard `GCN`, original `GATConv`, and `TransformerConv` layers. These changes resulted in comparable F1 scores—0.6901 for `GATConv` and 0.6960 for `TransformerConv`—indicating that the specific graph convolution operator does not significantly affect the final performance.

## 11 CONCLUSION

In this paper, we introduce Knowledge Debugger, a GNN-based framework for detecting inconsistencies in multimodal and structured content. By modeling knowledge as graphs, Knowledge Debugger surpasses traditional similarity-based and RAG approaches, achieving 11% higher F1 with lower retrieval cost on a benchmark built from Wikipedia and research papers. This work advances reliable knowledge systems and paves the way for applications in domains such as autonomous vehicles, healthcare, and observability systems.

REPRODUCIBILITY STATEMENT

We have taken several steps to ensure reproducibility of our work. The construction of the MKDB dataset is documented in Section 6 and Appendix B, with detailed preprocessing pipelines and synthetic perturbation procedures further described in Appendix B. Model architectures, hyperparameters, and training settings are provided in Appendix D. Comprehensive evaluation metrics and task definitions are given in Section 5, Appendix C and Appendix D. We will release anonymous source code and scripts as supplementary material to reproduce dataset construction, training, and evaluation. All baseline models and encoders are either open-source or referenced with precise versions, and random seeds are fixed for experiments where applicable.

ETHICS STATEMENT

This work develops Knowledge Debugger (Knowledge Debugger) and the Multimodal Knowledge Debugging Benchmark (MKDB) to study the detection and correction of inconsistencies in multimodal knowledge graphs. All data come from publicly available sources: Wikipedia (CC BY-SA 4.0) and arXiv papers released under a non-commercial license (CC BY-NC). We do not process sensitive personal data or involve human subjects, and no additional ethical approval was required.

We recognize potential risks, including misuse of inconsistency detection to generate more convincing fabricated content, as well as biases stemming from English-centric Wikipedia and computer science arXiv corpora. To mitigate these risks, we (i) clearly separate synthetic perturbations from original data, (ii) release provenance and usage guidelines, (iii) encourage human-in-the-loop review, and (iv) recommend evaluation across diverse domains. Our model is lightweight (<10M parameters), reducing environmental impact compared to large-scale LLM baselines.

Overall, this research aims to advance the trustworthiness and auditability of knowledge-intensive systems while being mindful of limitations and dual-use concerns.

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

## A    THE USE OF LARGE LANGUAGE MODELS (LLMS)

We used ChatGPT as a writing assistant to help us write part of the paper. Additionally, we utilize the power of CodePilot to help us code faster. However, all the AI-generated writing and coding components are manually checked and modified. There is no full AI-generated content in the paper.

## B    ASSETS DETAILS

### B.1    CODE AND DATA OPEN-SOURCE

We release our MKDB benchmark for knowledge debugging anonymously via `https://osf.io/df92q/?view_only=3657b0d249be40ef9754f4c46717e283`. We will release the codebase and dataset once our paper gets accepted.

### B.2    DATASET INFORMATION

Details about the benchmark corpus statistics, collection process, construction of debugging tasks, and evaluation metrics we rely on are available in Section 6 in the main content. In this section, we provide more technical details about dataset splitting and dataset construction prompts.

**Dataset split**    For node-level tasks, the dataset includes a total of 4,987 tasks for Wikipedia and 4,567 tasks for research papers. Each task contains one positive and one negative node. For edge-level tasks, we include 4,381 tasks for Wikipedia and 5,003 tasks for research papers. Each task includes one positive and one negative edge. For bug detection tasks at both node-level and edge-level, we allocate 200 tasks (200 positive examples, 200 negative examples for 400 classification tasks) as the test set, with the remaining used for training. For bug correction tasks at both node-level and edge-level, following the same protocol, 400 tasks that are randomly selected from the dataset (400 rewired edges for 400 retrieval tasks) are reserved for testing, and the rest are used for training.

**Dataset construction prompt**    For node-level task generation, we rely on prompting GPT-4o to construct false examples of the nodes. As shown in Table 4, we utilize such a prompt to rewrite the original content of one target node with its sampled neighborhoods.

| Role | Content |
| --- | --- |
| System | You are a creative research assistant. |
|  | Task: |
|  | 1. For each of the first $k$ source texts, invent exactly one false but plausible statement. |
|  | 2. Then rewrite the final text so that it coherently combines in all of those false statements. |
|  | Output *only* valid JSON with two fields: |
|  | • "fabrications": an array of the $k$ false statements, in order |
|  | • "final_node": the rewritten version of the final text |
| User | Here are the node texts: |
|  | 1. Text 1: "⟨first prefix text⟩" |
|  | 2. Text 2: "⟨second prefix text⟩" |
|  | . . . |
|  | $k$+1. Final text that needs to be rewritten $k + 1$: "⟨final text⟩" |
|  | Please perform the task as described. |

Table 4: Full chat prompt for fabricating and rewriting node texts

### B.3    DATASET LICENSE

The dataset comprises two components: research papers and Wikipedia content. To ensure the dataset is used solely for non-commercial purposes, we will release the research paper portion under the

Creative Commons Attribution-NonCommercial (CC BY-NC) license. For the Wikipedia portion, our data is sampled from the WikiWeb2M dataset, which was originally released under the Creative Commons Attribution-ShareAlike 3.0 Unported (CC BY-SA 3.0) license. In compliance with this, we will retain the same license for the Wikipedia-derived content in our dataset.

### B.4 MODEL INFORMATION

We utilize multiple models during our experiments, including text-based and multimodal retriever models (`all-MiniLM-L6-v2`, `voyage-multimodal-3`, `BAAI/AltCLIP`), LLM-based generation models (`Qwen/Qwen2.5-7B-Instruct-Turbo`), and GNN models (`GATConv2`). Typically, for `all-MiniLM-L6-v2`, we rely on the package of `sentence-transformers` for usage. For `BAAI/AltCLIP`, we utilize the checkpoint on Huggingface and utilize `transformers` package for inference. For `voyage-multimodal-3`, and `Qwen/Qwen2.5-7B-Instruct-Turbo`, we rely on the API of `voyage-ai` and `together-ai` for inference calling. For `GATConv2`, we rely on the implementation of `PyG` for usage, and we train from scratch without relying on existing model checkpoints.

### B.5 MODEL LICENSE

`Qwen2.5-7B-Instruct-Turbo`: Apache 2.0 License
`voyage-multimodal-3`: close-source, no available license

## C BENCHMARK CONSTRUCTION DETAILS

We build the benchmark in four deterministic stages:

1. **Collect** raw documents from Wikipedia and research papers.
2. **Parse** each document into nodes (paragraphs, images, captions) using rule-based procedures.
3. **Connect** reading-flow edges (`follow`, `reference`) according to document order.
4. **Link** structure by connecting sections and subsections to their corresponding content using the document hierarchy.

No content is altered; the graph can be losslessly reverted to the original material. The entire preprocessing is rule-based—no manual annotation or external supervision is required. Because Wikipedia articles and research papers are generally high-quality and knowledge-dense, we expect few inherent inconsistencies in the sources.

## D EXPERIMENTAL DETAILS

### D.1 COMPUTING RESOURCE

Since we focus on efficient training and usage of RAG and GNN experiments, all of our experiments on training and inference rely on one single A100 80GB GPU.

### D.2 PROMPT DETAILS FOR LLM BASELINES

The prompt for the LLM baseline is structured as follows:

Context: context

Query: Is the sentence consistent with other information in the relevant paragraph?

Sentence: sentence

Output: Return a JSON object in the format: "confidence": 1-5, "answer": "True" | "False" Context: The entire paper, with inconsistent sentences embedded.

The sentence is the target sentence under evaluation. And the output is a binary decision (True/False) with a confidence score from 1 (low) to 5 (high).

This setup ensures the model evaluates each candidate sentence in the context of the full paper. A sentence is consistent if it does not conflict with any part of the context, and inconsistent where there is at least one conflicting part.

### D.3    NODE-LEVEL BASELINES

Since node-level baselines do not require training, we exclude all training data for baseline results and focus on the test set.

**Bug correction task**    For the bug correction task, We evaluated four different retrievers to select the top-$k$ most relevant nodes using the fabrication content for each node:

- **Structure Retriever**: retrieves the top-$k$ nodes based on Personalized PageRank (PPR) scores computed from the seed node.

- **Text Retriever**: retrieves the top-$k$ nodes based on cosine similarity with the embeddings provided by `all-MiniLM-L6-v2`.

- **Multimodal Retriever**: retrieves the top-$k$ nodes based on cosine similarity with the embeddings provided by `voyage-multimodal-3` model to embed for the Wiki graph and `BAAI/AltCLIP` model to embed for the Paper graph.

- **Hybrid Retriever**: combines the scores of the text retriever and the structure retriever with weights of 0.8 and 0.2 linearly in the Wiki graph, with weights of 0.2 and 0.8 linearly in the Paper Graph, and selects the top-$k$ results as the final retrieved ones. Different combination values are selected for different datasets due the the different properties of graphs.

We reported the mean reciprocal rank (MRR), mean average precision (MAP), recall at $k = 5$, and normalized discounted cumulative gain (NDCG) at $k = 5$.

**Bug detection task**    For the bug detection task, we generate three prompt variants: *original prompt*, *oracle prompt*, and *Retrieval-Augmented Generation (RAG) prompt*. The *original prompt* is used for LLM-based baseline, and the *oracle prompt* is used for testing the upper-bound of the LLM-based method when given the ground-truth retrieved contents. We use the `Qwen/Qwen2.5-7B-Instruct-Turbo` to generate the final answer based on the retrieved information. Table 5 compares these templates. We reported the precision, recall, F1, and set the $k$ of the retrieval process to be 5. The retriever used as part of the RAG pipeline for bug detection is the same as the bug correction tasks.

| Original Prompt | Oracle Prompt | RAG Prompt |
|---|---|---|
| You are given a statement. Based on your knowledge, evaluate if this statement contains any factual errors. Statement: . . . Respond with 'Yes' if it does, otherwise 'No'. Just say Yes or No. Do not add any other text. | You are given a statement and multiple contents. Based on your knowledge, evaluate if this statement contains any very clear factual contradictary errors when comparing with at least one of the multiple contents, response Yes, else No. Statement: . . . Multiple contents: . . . Respond with 'Yes' if it does, otherwise 'No'. Just say Yes or No. Do not add any other text. | You are given a statement and multiple contents. Based on your knowledge, evaluate if this statement contains any very clear factual contradictary errors when comparing with at least one of the multiple contents, response Yes, else No. Statement: . . . Multiple contents: . . . Respond with 'Yes' if it does, otherwise 'No'. Just say Yes or No. Do not add any other text. |

Table 5: **Comparison of the original, oracle, and Retrieval-Augmented Generation (RAG) prompt templates.** The multiple contents in the oracle prompt are the node-neighbors' contents. The multiple contents in the RAG prompt are the retrieval contents.

### D.4  Edge-level Baselines

Similar with node-level baselines, since edge-level baselines do not require training, we exclude all training data for baseline results and directly use the test set.

**Bug correction task**  For the rewired-neighbor edges, we removed the original edge and inserted the rewired edge to form a modified graph. We evaluated four different retrievers to select the top-$k$ most relevant nodes, excluding nodes of type `image`. The retriever settings are similar to the node-level tasks. We reported the mean reciprocal rank (MRR), mean average precision (MAP), recall at $k = 5$, and normalized discounted cumulative gain (NDCG) at $k = 5$.

**Bug detection task**  For the bug detection task, we keep the 200 edge-pairs (200 positive and 200 negative) for testing. For the original-neighbor edges, we retrieve in the original graph. For the rewired-neighbor edges, we retrieve them in the modified graph as in the bug correction task. Then, we retrieve it by following the same process. For each edge, we generated two prompt variants: the *original prompt* and the *Retrieval-Augmented Generation (RAG) prompt*. The *original prompt* is used as the LLM baseline. We use the `Qwen/Qwen2.5-7B-Instruct-Turbo` to generate the answer. Table 6 compares these templates. We report the precision, recall, and F1 and take the retrieval number $k = 5$.

| **Original Prompt** | **RAG Prompt** |
|---|---|
| You are given two paragraphs. 
 Paragraph 1: ... 
 Paragraph 2: ... 
 Based on the background information in the reference paragraphs, do you think these two paragraphs are closely related? 
 Respond with 'Yes' or 'No'. | You are given two paragraphs and multiple reference paragraphs. 
 Paragraph 1: ... 
 Paragraph 2: ... 
 References for Paragraph 1: ... 
 References for Paragraph 2: ... 
 Based on the background information in the reference paragraphs, do you think these two paragraphs are closely related? 
 Respond with 'Yes' or 'No'. |

Table 6: **Comparison of the original and Retrieval-Augmented Generation (RAG) prompt templates.** The original prompt is the baseline while the RAG is the augmented one.

### D.5  GNN Training Details

We filter the training data by guaranteeing that the sub-graph we consider during training and testing does not overlap to avoid any form of data leakage. Our GNN training includes a batch size of 16, a GNN with 3 layers, each having 128 hidden units and 4 attention heads, a dropout rate of 0.2 between layers, an AdamW optimizer with a learning rate of $1 \times 10^{-3}$, and 20 training epochs.

### D.6  Significant Test

We run GNN-based training three times, and the performance indicates that our GNN-based models are significantly better than baselines on node-level and edge-level bug detection and edge-level bug correction, but not significantly better on edge-level bug detection.

## E  Broader Impact

This paper introduces the knowledge debugger, a tool designed to detect multimodal automatically information inconsistencies in real-world, knowledge-intensive materials such as research papers and Wikipedia pages. Our method aims to improve the quality of factual content and reduce the prevalence of incorrect information. However, if misused, it could also be exploited to generate highly convincing fake content that is difficult to distinguish from authentic sources, potentially contributing to the spread of misinformation.

# F LIMITATIONS

Our proposed knowledge debugger has several limitations. First, it focuses on only two representative types of source material—research papers and Wikipedia pages—for graph construction and debugging analysis. Expanding to a broader range of structured materials, such as books, films, and code repositories could enhance its generality. Second, our current task setting is limited to detecting knowledge inconsistencies. Identifying an inconsistency does not necessarily imply the ability to determine which source is correct. Finally, our bug correction task is formulated as a ranking problem: successfully locating the source of an error does not guarantee the ability to correct it.

