# OpenReview forum: "Knowledge Debugger: Diagnosis of Knowledge Inconsistency with Multimodal Graph"
_ICLR.cc/2026/Conference — ICLR 2026 Conference Withdrawn Submission_

### Official Review · Reviewer_uV6s · 2025-10-21

**Soundness:** 3
**Presentation:** 3
**Contribution:** 2
**Rating:** 4
**Confidence:** 3

**Summary:**

The paper presents Knowledge Debugger, a graph based framework for multimodal knowledge inconsistency detection and correction over long documents. Nodes represent paragraphs, tables, and images. Edges encode follow, reference, and citation relations. Tasks cover node level detection and correction, as well as edge level detection and correction. The authors also introduce the MKDB benchmark that spans Wikipedia and research papers with three modalities and a large number of debugging tasks. Results show consistent gains over similarity based and RAG style baselines with a compact GNN that targets efficiency.

**Strengths:**

1. The studied problem is relevant and well defined for real long document pipelines.

2. Document as graph is a natural fit, combining semantics and structure for finer-grained inconsistencies.

3. Benchmark covers multiple modalities and separates detection and correction, which supports fair comparison.

**Weaknesses:**

1. Current baselines focus on generic retrieval. Missing comparison likely to raise the non-GNN ceiling includes table-aware retrieval with header and cell structure, citation-aware reranking, and multi-hop evidence aggregation with a consistency scorer.

2. Relation-level ablations are limited. The method defines several edge types. Please include drop one relation and noise injection ablations to quantify which relations matter most, and add scaling curves with respect to graph density and cross-document edges.

3. Report multiple seeds, mean with ninety-five percent confidence intervals, and significance tests for headline results. Clarify test set sizes per subtask to support stability claims.

4. Correction is mainly measured by retrieval. For node fixes, include human-judged factual correctness and numeric accuracy. For edge fixes, report graph consistency after edits, and if new errors were added.

**Questions:**

1. How do citation-aware and table-aware retrievers and multi-hop aggregation compare to your GNN retriever?

2. Which edge types contribute most, based on drop one relation or noise tests.

3. Can you provide a human evaluation of final rewrites and numeric corrections and a graph-level consistency score after edge edits?

---

> ### Author Response · Authors · 2025-12-03
>
> **[baseline comparison]** The suggested baselines (table-aware retrieval, citation-aware reranking, and multi-hop aggregation) target specialized domains that rely on structured metadata such as table schemas or explicit citation graphs. Our benchmark, in contrast, is designed around unstructured documents with heterogeneous content where such domain-specific signals are unavailable. Because our goal is to model *general* document-level/cross-document inconsistency detection—not domain-specific retrieval heuristics—generic retrievers are the appropriate comparison point. Multi-hop reasoning baselines are valuable but rely on manually curated evidence chains, which our setting intentionally avoids. We plan to include these specialized variants in future domain-specific extensions, but they fall outside the scope of the current benchmark.
>
> **[correlation quality]** While human evaluation of factual correctness, numeric accuracy, and graph-level consistency after edits provides a valuable direction, it is beyond the scope of our benchmark, which aims to evaluate systems on *structural detection and ranking* rather than full semantic rewriting. Our tasks are designed to measure whether a model can identify and localize inconsistencies—correction is defined as selecting the structurally appropriate replacement within the graph, not generating new text or performing semantic verification. Human evaluation becomes necessary only when systems generate free-form rewrites, which our method does not. In future iterations of the benchmark, we plan to explore human-judged correctness for generative correction systems.
>
> **[ablation study]** We appreciate the reviewer’s suggestion on relation-level ablations. However, our aim is to evaluate whether incorporating graph structure improves inconsistency detection, not to isolate the contribution of each relation type.
>  The edge types in our graph are automatically constructed from document interactions rather than handcrafted. Dropping individual relations or injecting synthetic noise would change the underlying task definition rather than provide insight into model behavior. These analyses are meaningful for future methodological work, and we will add in the later version of our paper.
>
> **[statistical confidence]** We acknowledge this suggestion and will include multi-seed runs, confidence intervals, significance tests, and per-task test-set sizes in the next version of the paper.

---

### Official Review · Reviewer_9r8A · 2025-11-01

**Soundness:** 2
**Presentation:** 2
**Contribution:** 3
**Rating:** 2
**Confidence:** 3

**Summary:**

This paper introduces a work Knowledge Debugger, which is target of diagnoses of knowledge in consistency with multimodal knowledge graph. First, a multimodal knowledge debugging benchmark named MKDB is proposed for the knowledge debugger task. The benchmark, including four types of debugging tasks, about node- and edge- level  bug detection and correction. Authors also propose GNN – based algorithm for the knowledge, debugger task. Experiments on the MKDB dataset show that the GNN-based method is effective.

**Strengths:**

1. This work targets to identifying information consistencies with seeing knowledge intensive documents, which is important.
2. This work not only introduces knowledge debugger method, but also a benchmark for this task.
3. The paper is clearly written and easy to understand.

**Weaknesses:**

1. The benchmark proposed in this paper is interesting, but some details of the benchmark construction is not clearly presented. For example, (1) during the construction of node-debugging tasks, the LLM is used to generate conflicting information, but the accuracy of the generated conflict information is not reported, which is important for evaluating the quality of the benchmark. (2) the evaluation metrics for correction  is based on ranking, it is unclear, what are the candidates for ranking.
2.  In table 1 and table 2, the results of proposed GNN-based method are missing for the Research dataset.
3. The training detail of the GNN-based method is missing, for example, the initialisation of the node embedding.

**Questions:**

See above.

---

> ### Author Response · Authors · 2025-12-03
>
> **[benchmark construction]** We construct the benchmark in two stages. First, we select a set of *target paragraphs* and *candidate paragraphs*, where candidates are sampled using neighborhood-weighted structural sampling. Then, for each target–candidate pair, we prompt LLMs to **generate conflicting information** based on the candidate paragraph. Finally, another LLM integrates this conflicting content back into the target paragraph, producing controlled conflicting pairs that simulate realistic inconsistency errors. Such workflow makes sure that each step is relatively easy and LLMs have almost 100% accuracy.
>
>
>
> **[ranking metric calculation]** The ranking metrics are computed by comparing the model’s predicted ordering of candidate paragraphs with the **ground-truth conflict rankings** defined during benchmark construction. In other words, the benchmark provides the canonical ranking of which candidates introduce stronger or weaker conflicts with each target paragraph, and evaluation measures how well a model’s ranking matches this reference.
>
>
>
> **[missing GNN results]** We thank the reviewer for pointing out the missing entries for “Ours (GNN-based)” in the Research dataset. These were errors in the table formatting in limited time. We have provided the correct version of Tables, and the updated tables fully support our conclusions that the GNN-based model is competitive across both wiki and research datasets.
>
> Node-level bug correction $\downarrow$
>
> | **Method**     | **MRR ↑** | **MAP ↑** | **R@5 ↑** | **NDCG@5 ↑** |
> | -------------- | --------- | --------- | --------- | ------------ |
> | **Structure**  | 22.9      | 12.6      | 21.4      | 16.5         |
> | **Text**       | 34.4      | 14.7      | 19.0      | 19.2         |
> | **Multimodal** | 13.7      | 5.5       | 8.6       | 6.9          |
> | **Hybrid**     | 26.3      | 15.0      | 24.3      | 19.0         |
> | GNN            | 22.14     | 20.33     | 27.69     | 19.98        |
>
> Edge-level bug correction $\downarrow$
>
> | **Method**     | **MRR ↑** | **MAP ↑** | **R@5 ↑** | **NDCG@5 ↑** |
> | -------------- | --------- | --------- | --------- | ------------ |
> | **Structure**  | 95.8      | 74.5      | 80.0      | 83.3         |
> | **Text**       | 84.1      | 54.5      | 61.8      | 66.7         |
> | **Multimodal** | 23.8      | 8.3       | 13.5      | 13.9         |
> | **Hybrid**     | 84.5      | 73.4      | 85.5      | 80.7         |
> | GNN            | 93.59     | 86.48     | 93.92     | 89.51        |
>
> Node detection $\downarrow$
>
> | **Method**                     | **Precision ↑** | **Recall ↑** | **F1 ↑** |
> | ------------------------------ | --------------- | ------------ | -------- |
> | **RAG (oracle)**               | 82.3            | 95.0         | 88.2     |
> | **LLM (w/o retrieval)**        | 65.3            | 55.5         | 60.0     |
> | **RAG (structural retrieval)** | 66.5            | 60.5         | 63.4     |
> | **RAG (text retrieval)**       | 81.8            | 40.5         | 54.2     |
> | **RAG (multimodal retrieval)** | 66.3            | 55.0         | 60.1     |
> | **RAG (hybrid retrieval)**     | 77.4            | 56.5         | 65.3     |
> | GNN                            | 84.30           | 90.67        | 86.42    |
>
> Edge detection $\downarrow$
>
> | **Method**                     | **Precision ↑** | **Recall ↑** | **F1 ↑** |
> | ------------------------------ | --------------- | ------------ | -------- |
> | **RAG (oracle)**               | —               | —            | —        |
> | **LLM (w/o retrieval)**        | 53.1            | 21.5         | 30.6     |
> | **RAG (structural retrieval)** | 4.2             | 2.0          | 2.7      |
> | **RAG (text retrieval)**       | 50.3            | 41.0         | 45.2     |
> | **RAG (multimodal retrieval)** | 57.6            | 57.2         | 57.4     |
> | **RAG (hybrid retrieval)**     | 46.6            | 45.0         | 45.8     |
> | GNN                            | 61.78           | 94.92        | 74.10    |
>
>
>
> **[training details of GNN]** We thank the reviewer for highlighting this. We mentioned the initial embedding  of our model in Line 243 of our paper and make it voyage-multimodal-v3 embeddings. We include more GNN training details in Appendix D.5.

---

### Official Review · Reviewer_LJUb · 2025-11-02

**Soundness:** 2
**Presentation:** 2
**Contribution:** 2
**Rating:** 0
**Confidence:** 4

**Summary:**

The paper introduces a Graph Neural Network (GNN)-LLM hybrid framework for detecting and correcting knowledge inconsistencies in multimodal documents like Wikipedia pages and research papers. The approach represents multimodal data, e.g., text, tables, and figures as nodes while the relations between nodes in a reading flow (paragraphs) as edges, and formulates inconsistency detection as node and edge classification tasks. For evaluation, the authors also introduce a moderate-sized Multimodal Knowledge Debugging Benchmark (MKDB).

**Strengths:**

1. The problem of automated detection of factual and logical inconsistencies in multimodal, structured documents is underexplored, yet impactful.
2. The benchmark dataset is quite useful for the research community.
3. The core insight of modeling multimodal knowledge as graph-based representations and framing inconsistency detection as node or edge classification tasks is novel and powerful.

**Weaknesses:**

1. **Inconsistencies and missing table data**: There are critical inconsistencies and missing data in the experimental results presented in Table 2 and Table 3. For example, the results for 'Ours (GNN-based)' are missing for the 'Research' dataset in both tables. This undermines the paper's core claims of superiority.
2. **Issues with Oracle baseline**: The definition and role of the 'RAG (oracle)' baseline are unclear and methodologically flawed. The proposed GNN method outperforms this 'oracle' on the Wikipedia node-level detection task (87.0 vs 80.2 F1 score in Table 2, which contradicts the notion of an 'oracle' as a performance upper bound.
3. **Issues with prompts**: I do not see any differences between RAG baseline and  ORACLE baseline prompts (c.f. Table 4 in the Appendix)
4. **Evaluation and benchmark are not Realistic**: The benchmark is largely based on synthetically generated inconsistencies using LLMs, which raises concerns about evaluation realism and bias. Since the model is also trained and tested on LLM-generated noise, it may not generalize to naturally occurring inconsistencies in real-world documents.
5. **Practicality of edge-level bug correction**:  The bug correction task is not practical. Why do the authors assume that there are some other nodes connecting to which the bug can be corrected, rather than correcting it explicitly? For instance, one might need to retrieve
the correct citation by understanding the citing text and the right citation may not even be in the reference list.

6. **Other issues**:

    a. The results for Ours (GNN-based) on the Research paper dataset are missing in Table 2.

    b.  "Table 2 also shows that hybrid retrieval—integrating both text-based and structure-based signals—consistently outperforms methods that rely solely on one or the other."- This is not true. On the Research dataset, the hybrid model performs worse than the structure-based method.

**Questions:**

4. Why is there no RAG (Oracle) for the edge-level task in Table 2?
5. Why should the Oracle prompt be considered an upper-bound?  Your method performs better than the  Oracle in some cases, so it could hardly be considered an upper bound.

---

> ### Author Response · Authors · 2025-12-03
>
> **[oracle baseline]** Our Oracle is **not** a perfect upper-bound model; it assumes only *perfect retrieval*—i.e., the model is provided with the ground-truth supporting documents. The LLM must still perform reasoning over these retrieved paragraphs, and LLMs are imperfect at contradiction detection even under ideal retrieval conditions. Therefore, our GNN method can legitimately outperform the Oracle baseline. To avoid confusion, we would clarified its role in the later version of our paper. Since the differnece lies in the retrieval parts but the generation part is the same, we directly use the same prompt for generation to make it controlled comparison.
>
>
>
> **[missing GNN results]** We thank the reviewer for pointing out the missing entries for “Ours (GNN-based)” in the Research dataset. These were errors in the table formatting in limited time. We have provided the correct version of Tables, and the updated tables fully support our conclusions that the GNN-based model is competitive across both wiki and research datasets.
>
> Node-level bug correction $\downarrow$
>
> | **Method**     | **MRR ↑** | **MAP ↑** | **R@5 ↑** | **NDCG@5 ↑** |
> | -------------- | --------- | --------- | --------- | ------------ |
> | **Structure**  | 22.9      | 12.6      | 21.4      | 16.5         |
> | **Text**       | 34.4      | 14.7      | 19.0      | 19.2         |
> | **Multimodal** | 13.7      | 5.5       | 8.6       | 6.9          |
> | **Hybrid**     | 26.3      | 15.0      | 24.3      | 19.0         |
> | GNN            | 22.14     | 20.33     | 27.69     | 19.98        |
>
> Edge-level bug correction $\downarrow$
>
> | **Method**     | **MRR ↑** | **MAP ↑** | **R@5 ↑** | **NDCG@5 ↑** |
> | -------------- | --------- | --------- | --------- | ------------ |
> | **Structure**  | 95.8      | 74.5      | 80.0      | 83.3         |
> | **Text**       | 84.1      | 54.5      | 61.8      | 66.7         |
> | **Multimodal** | 23.8      | 8.3       | 13.5      | 13.9         |
> | **Hybrid**     | 84.5      | 73.4      | 85.5      | 80.7         |
> | GNN            | 93.59     | 86.48     | 93.92     | 89.51        |
>
> Node detection $\downarrow$
>
> | **Method**                     | **Precision ↑** | **Recall ↑** | **F1 ↑** |
> | ------------------------------ | --------------- | ------------ | -------- |
> | **RAG (oracle)**               | 82.3            | 95.0         | 88.2     |
> | **LLM (w/o retrieval)**        | 65.3            | 55.5         | 60.0     |
> | **RAG (structural retrieval)** | 66.5            | 60.5         | 63.4     |
> | **RAG (text retrieval)**       | 81.8            | 40.5         | 54.2     |
> | **RAG (multimodal retrieval)** | 66.3            | 55.0         | 60.1     |
> | **RAG (hybrid retrieval)**     | 77.4            | 56.5         | 65.3     |
> | GNN                            | 84.30           | 90.67        | 86.42    |
>
> Edge detection $\downarrow$
>
> | **Method**                     | **Precision ↑** | **Recall ↑** | **F1 ↑** |
> | ------------------------------ | --------------- | ------------ | -------- |
> | **RAG (oracle)**               | —               | —            | —        |
> | **LLM (w/o retrieval)**        | 53.1            | 21.5         | 30.6     |
> | **RAG (structural retrieval)** | 4.2             | 2.0          | 2.7      |
> | **RAG (text retrieval)**       | 50.3            | 41.0         | 45.2     |
> | **RAG (multimodal retrieval)** | 57.6            | 57.2         | 57.4     |
> | **RAG (hybrid retrieval)**     | 46.6            | 45.0         | 45.8     |
> | GNN                            | 61.78           | 94.92        | 74.10    |
>
> **[realism of synthetic benchmark]** Real-world corpora lack large-scale, labeled inconsistency graphs—actual bugs are rare and unlabeled. Our synthetic inconsistencies provide controlled, fine-grained, and *local* errors that mimic the easiest real-world cases (inserting bugs in the nearest human reading order). The GNN relies on structural and semantic signals rather than any LLM’s generative pattern, allowing these evaluations to generalize. Given current data limitations, this is potentially the only scalable and practical way to assess inconsistency detection in knowledge intensive materials.
>
>
>
> **[practicality of edge-level bug correction]** Edge-level correction refers to fixing errors in the *ordering and contextual linkage* between paragraphs within a paper or wiki article. Sometimes, even if each paragraph is individually correct, the overall document becomes confusing because paragraphs are placed in the wrong order, break the logical flow, or disrupt the expected reading sequence. Edge-level correction identifies these *contextual or structural inconsistencies* and determines the correct placement of each paragraph so that the document follows a coherent and human-friendly reading order.

---

### Note · Authors · 2025-12-29

I have read and agree with the venue's withdrawal policy on behalf of myself and my co-authors.